# Molecular and Cellular Risk Assessment of Healthy Human Cells and Cancer Human Cells Exposed to Nanoparticles

**DOI:** 10.3390/ijms21010230

**Published:** 2019-12-28

**Authors:** Edward Helal-Neto, Aline Oliveira da Silva de Barros, Roberta Saldanha-Gama, Renata Brandão-Costa, Luciana Magalhães Rebêlo Alencar, Clenilton Costa dos Santos, Ramón Martínez-Máñez, Eduardo Ricci-Junior, Frank Alexis, Verônica Morandi, Christina Barja-Fidalgo, Ralph Santos-Oliveira

**Affiliations:** 1Brazilian Nuclear Energy Commission, Nuclear Engineering Institute, Rio de Janeiro 21941906, Brazil; edwneto@gmail.com (E.H.-N.); alinedbcg@gmail.com (A.O.d.S.d.B.); 2Laboratory of Cellular and Molecular Pharmacology, Department of Cell Biology, IBRAG, Universidade do Estado do Rio de Janeiro, Rio de Janeiro-RJ 21040900, Brazil; saldanhagama@yahoo.com.br (R.S.-G.); entaabrandaocosta@gmail.com (R.B.-C.); cbarja.uerj@gmail.com (C.B.-F.); 3Department of Physics, Federal University of Maranhão, São Luis do Maranhão 65080-805, Brazil; lucianamagal@gmail.com (L.M.R.A.); cleniltoncs@gmail.com (C.C.d.S.); 4Instituto Interuniversitario de Investigación de Reconocimiento Molecular y Desarrollo Tecnológico (IDM), Universitat Politècnica de València, Universitat de València. Camino de Vera s/n, 46022 Valencia, Spain; rmaez@qim.upv.es; 5Departamento de Química, Universidad Politécnica de Valencia, Camino de Vera s/n, 46022 Valencia, Spain; 6CIBER de Bioingeniería, Biomateriales y Nanomedicina (CIBER-BBN), Jordi Girona 18-26, ES 08034 Barcelona, Spain; 7Faculty of Pharmacy, Federal University of Rio de Janeiro, Rio de Janeiro-RJ 21941-901, Brazil; ricci@pharma.ufrj.br; 8Department of Bioengineering, Clemson University, Clemson, SC 29634, USA; falexis@yachaytech.edu.ec; 9Laboratory of Biology of Endothelial Cells and Angiogenesis (LabAngio), Department of Cell Biology, IBRAG, Universidade do Estado do Rio de Janeiro, Rio de Janeiro-RJ 20550-900, Brazil; morandi.v@gmail.com; 10Laboratory of Radiopharmacy and Nanoradiopharmaceuticals, Zona Oeste State University, Rio de Janeiro-RJ 23070-200, Brazil

**Keywords:** nanoparticles, nanomedicine, tumor cell, cell culture, primary cells

## Abstract

Nanodrugs have in recent years been a subject of great debate. In 2017 alone, almost 50 nanodrugs were approved for clinical use worldwide. Despite the advantages related to nanodrugs/nanomedicine, there is still a lack of information regarding the biological safety, as the real behavior of these nanodrugs in the body. In order to better understand these aspects, in this study, we evaluated the effect of polylactic acid (PLA) nanoparticles (NPs) and magnetic core mesoporous silica nanoparticles (MMSN), of 1000 nm and 50 nm, respectively, on human cells. In this direction we evaluated the cell cycle, cytochemistry, proliferation and tubulogenesis on tumor cells lines: from melanoma (MV3), breast cancer (MCF-7, MDA-MB-213), glioma (U373MG), prostate (PC3), gastric (AGS) and colon adenocarcinoma (HT-29) and non-tumor cell lines: from human melanocyte (NGM), fibroblast (FGH) and endothelial (HUVEC), respectively. The data showed that an acute exposure to both, polymeric nanoparticles or MMSN, did not show any relevant toxic effects on neither tumor cells nor non-tumor cells, suggesting that although nanodrugs may present unrevealed aspects, under acute exposition to human cells they are harmless.

## 1. Introduction

A great variety of new nanoparticles (NPs), with specific features have been under intense investigation [1,2,3,4,5,6,7,8]. Also, there are many studies on the effects of these nanoparticles within the human body [9,10,11,12], and a huge number of new nanodrugs have been approved for clinical use worldwide [13]. Despite all these data, and the supposed advantages of nanomedicine/nandodrugs, many aspects of the biological interaction as the human behavior are still unrevealed, especially in the molecular and cellular level. Some studies have shown that nanopolymers may induce cytotoxicity, inflammatory or immune response with reverberation in the biological systems. Other studies have shown that nanoparticles may produce reactive oxygen species (ROS) and free radicals, resulting in oxidative stress, inflammatory events, deoxyribonucleic acid (DNA) damage, multinuclear formation, and fibrosis [12,14,15,16]. In the last few years some studies have shown that nanoparticles are capable to pass thru important biological barriers as: blood–brain barrier [17,18,19,20,21] and transplacental barrier [22,23,24]. All these data suggest that many aspects regarding the nanoparticles effects on humans body, especially in molecular and cellular level are still unrevealed, and must be evaluated [25,26].

Among all types of nanoparticles studied worldwide, mesoporous silica and polymeric nanoparticles are the two that have been investigated more. Mesoporous silica nanoparticles (MSNs), have a high surface area, unvarying porosity, stable aqueous dispersion, and in vivo biodegradability [27]. In the last five years, the number of published data describing the use of MSN has increased significantly, especially for biomedical applications, where precise control over particle size, shape, pore size, and pore geometry is highly appreciated [28,29,30]. For example, Mou and co-workers showed that the cellular uptake of MSN in Hela cells increased 2.5 fold just varying the size from 50 nm to 30 nm [31]. Another study using spherical MSNs with sizes of 190, 420, and 1220 nm showed that the cytotoxicity is dependent on the size, by which smaller MSN (190 nm) are more cytotoxic than bigger MSN (1220 nm), due to the decreased endocytosis of bigger sizes [32]. Also, MSN with a cationic charge would induce more cytotoxicity than the neutral and anionic MSN, although the latter have a higher transvascular transport in tumors [33,34,35].

Polymeric nanoparticles are solid particles with a size in the range of 10–1000 nm. They allow encapsulation of the drugs inside a polymeric matrix, protecting them from enzymatic and hydrolytic degradation [36]. Nanocarriers with diameters below 200 nm are better at achieving extravasation into tumors by the enhanced permeability and retention (EPR) effect [37]. Biodegradable polymers are of particular interest because they can be fully metabolized and removed from the body [15]. Among all polymeric nanoparticles, the polylactic acid (PLA) nanoparticles are the most investigated and used. PLA nanoparticle delivery systems have been developed for therapy, diagnosis, and imaging. They can range in size from a single polymer chain— used directly as a therapeutic or as a modifying agent for a drug or diagnostic agent—to large aggregates within the nanoscale [38]. A doxorubicin-loaded (DOX-loaded) PLA nanoparticles followed by the conjugation of a Mn-porphyrin on the surface of the nanoparticles were sensitive to pH and inhibiting the growth of HeLa and colon adenocarcinoma (HT-29) cells [39]. PLA nanoparticles encapsulating docetaxel (DTX) showed a high specific-binding and anti-metastatic ability [40]. Also, polylactic acid (PLA)/polyvinyl alcohol(PVA)/montmorillonite (MMT)-trastuzumab (PLA/PVA/MMT/trastuzumab labeled with 99mTc was capable of reaching breast cancer cells [41].

In this study we evaluated the effect of two different nanoparticles: polylactic acid (PLA) polymeric nanoparticles and magnetic core mesoporous silica nanoparticles (MMSN), of 1000 nm and 50 nm, respectively, on healthy human cells. Thus, human tumor cells from melanoma (MV3), breast cancer (MCf-7 and MDA-MB-231), glioma (U373MG), prostate (PC3), gastric (AGS) and colon adenocarcinoma (HT-29) and non-tumor cells from human melanocyte (NGM), fibroblast (FGH) and endothelial (HUVEC) were exposed to acute doses of both nanoparticles (separately) and the cellular, morphological and molecular aspects of these cells were evaluated in depth.

## 2. Results and Discussion

### 2.1. Nanoparticles Fabrication and Characterization

#### 2.1.1. Magnetic Core Mesoporous Silica Nanoparticles (MMSN)—Small Nanoparticles (50 nm)

The PXRD performed in low angles confirmed the periodicity of the mesoporous material produced. It is important to notice the formation of the sharp peak in **S0-1** and in the calcined sample **S0-2.** The analysis performed in a high angle showed the condensation of silanol groups during the calcination process as the formation of magnetic core within the structure (inset in Figure 1). The mesoporous structure of **S0-2** magnetic core MSNs was also analyzed by transmission electron microscopy and the size of the primary nanoparticles was determined by image analysis (58.9 ± 8.1 nm, *n =* 100). The data was represented in a histogram, which shows the particle size distribution of the **S0-2** nanoparticles (Figure 1C). Finally, the N_2_ adsorption-desorption analysis confirmed the mesoporous material formation, showing a surface area of 872 m^2^/g, with a pore volume of 0.85 cm^3^/g and a pore diameter of 3.15 nm.

#### 2.1.2. Polylactic Acid (PLA) Polymeric Nanoparticles—Giant Nanoparticles (1000 nm)

The polylactic acid polymeric nanoparticles presented a mean size of 929.47 ± 37.72 nm, with a polydispersity index (PDI) of 0.228 ± 0.05 showing homogeneous size for the nanoparticles (Figure 2). The system showed a very low PDI, which indicates that this big nanoparticles also have a monodisperse behavior. The Raman spectroscopy analysis corroborated the spherical shape and composition of the microparticles.

It is possible to observe the uniformity of the composition of the microparticle based on the analysis varying on the z and y axis, which also corroborates the dynamic light scattering (DLS) data.

### 2.2. Effect of Nanoparticles on Tumor and Non-Tumor Cells

#### 2.2.1. Cell Viability—Proliferation

Nanoparticles may be designed for several applications, including imaging, therapy, and as theranostics to be used in a wide range of diseases, including oncology, cardiovascular, and neurology [42,43,44]. In this direction, the evaluation of non-loaded NPs is quite desirable in order to understand the real effect of these nanoparticles on the cellular, morphological and molecular aspect. In order to evaluate the cell viability we performed the MTT assay testing a dose of 20 ug/mL. This dose has been used by our group in a variety of studies in vitro [22,45,46]. However, there is a lack of evidence related to the toxic effects of this dose. Also, we chose this value in order to mimic a human dose. MTT readout is a measure of total metabolic activity in a cell culture. It can be altered by changes in cell cycle, size or survival. The data presented in Figure 3 shows that none of the NPs used showed any significant effect on tumor cell viability. The same result was observed in non-tumor cells line (Figure 4).

This results corroborates that the exposition of cells cultures (tumor an non-tumor) to non-loaded nanoparticles (polymeric and magnetic mesoporous silica) does not alters their viability, which means that using an acute dose (20 μg/mL) of each nanoparticle was not able to prevent or hinder cell growth.

#### 2.2.2. Cell Cycle

Despite the lack of effect in cell surviving, we decided to validate this result performing the cell cycle assay by quantitation of DNA. Cell cycle assay by quantitation of DNA content, allows the analyses of the percentage of cells in the different phases of the cell cycle (G1, S and G2) and the percentage of apoptotic cells (SubG1). The results (Figure 5) demonstrate that DNA content analyses did not show any effect of neither in non-tumor (fibroblast, endothelial and melanocyte) nor tumor cells.

This data, in accordance with the MTT assay, demonstrated that both nanoparticles are not capable to produce any distinct effect on cells line, and may corroborate the safety of these nanoparticles.

### 2.3. Cell Morphology

Polylactic acid polymeric nanoparticles and MMSN presents outstanding and specific physicochemical properties that may influence their interaction with target cells. However, these properties may also be responsible for toxic effects [47]. Size increases the entry of NPs into the cells but it also exerts a potential damaging action in sites barred from larger particles [48]. Spherical nanoparticles are less toxic than rod ones and are internalized faster when compared to NPs with different shapes [49]. During in vitro cell culture procedures, it is important that cells preserve their morphology throughout the entire time of culture. Typical signs of problems includes granularity around the nucleus, detachment of the cells from the substrate, and size variations. These signs may be consequence of contamination of the culture, cell senescence or the presence of toxic agents in the medium.

The results presented in Figure 6A,B showed that both, polylactic acid polymeric nanoparticles and MMSN, were capable to alter the granularity and size of tumor and non-tumor cells (24 post-incubation when just compared to untreated cells (controls). Is important to notice that more studies including nanoparticles-trafficking into the cells as the use of a positive control should be performed in order to deterministically establish the status and the mechanism of the morphological alteration found.

Another important morphological parameter is the rearrangement of the cytoskeleton. Changes in the dynamics of the actin microfilaments may result in different cell destiny. A loss of actin polymerization could lead to cell detachment which, ultimately, may result in cell death. However, actin microfilaments rearrangement can be associated with the activation of signaling pathways that regulate cell behavior [50]. Also, changes in actin polymerization can be a consequence of cell migration when an increase in the formation of stress fiber in the migrating cell is observed [51].

In this study, when human tumor cells were incubated with polylactic acid polymeric nanoparticles or MMSN, we did not observe any change in the actin cytoskeleton rearrangement (Figure 7) when compared with untreated groups (control). This lack of effect was also observed when non-tumor cells, HUVEC, fibroblast or melanocyte, were incubated polylactic acid polymeric nanoparticles or MMSN (Figure 8). Although we observed a slight decrease in the fluorescence level when MV3 cells were treated with MMSN (Appendix A), cell morphology and cytoskeleton reorganization remained similar to the control group.

### 2.4. Intracellular Signaling

Cell survival also relies on in the maintenance of intracellular pathways signaling. Akt phosphorylation, which has been shown to modulate migration, proliferation and survival of endothelial cells are the main pathways/connections that may be affected during cell migration and proliferation [52]. The MAPK/Erk signaling cascade is initiated by several receptors and regulates cellular growth and differentiation. Erk phosphorylation may activate targets in the cytosol and also translocate to the nucleus regulating gene expression [53]. Last, glyceraldehyde-3-phosphate dehydrogenase (GAPDH) is a glycolytic enzyme widely used as an internal control in experiments on proteins, mRNA, and DNA. However, emerging evidences indicates that GAPDH is implicated in diverse functions independent of its role in energy metabolism [54]. Levels of GAPDH are also deregulated in various cancer cells and also in normal cells, as endothelial cells [55,56,57].

In this study, when fibroblasts, HUVECs and melanocytes cells were treated with polylactic acid polymeric nanoparticles or MMSN, we did not observe changes in the phosphorylation of Akt and ERK1/2 as it did not modify GAPDH expression (Figure 9). Also, after treatment of tumor cells (MV3, MDA-MB-231, MCF-7, U373, PC-3 and HT-29) with polylactic acid polymeric nanoparticles or Magnetic Core Mesoporous Silica (MMSN), we did not observe any effect in the Akt and ERK1/2 phosphorylation as in GAPDH expression as demonstrated in Figure 10.

It is important to notice, however, that the AGS cells, when exposed to both, polylactic acid polymeric nanoparticles (20 ug/mL), or magnetic core mesoporous silica MMSN (20 ug/mL) increased in the expression of Akt phosphorylation. The group of AGS cells exposed to MMSN also showed an increased GAPDH expression. This set of data showed that with the exception of AGS cells, NPs does not have any significant effect in these signaling pathways. However, more studies should be done in order to better understanding understand the mechanism involved in this case.

### 2.5. Cell Migration and Tubulogenesis

To have a better understanding about the toxic effects of the acute exposure of NPs on cells, we treated non-tumor and tumor cells with polylactic acid polymeric nanoparticles or MMSN for 24 h and analyzed cell migration.

The results are expressed in Figure 11. Data showed that NPs did not have any effect in on cell migration of MV3, MDA-MB-231 and HT-29 (Figure 11A,B,G) cells. However, polylactic acid polymeric nanoparticles increased the wound closure of PC3 (39%) cells (Figure 11E) when compared to the untreated group (controls). This suggests that PLA NPs may induce cellular migration in this tumor cell line. Similarly, when MCF-7 and AGS cells were treated with MMSN, it increased the cell migration (58% and 53%, respectively) when compared to the control group (Figure 11C,F). On the other hand, MMSN reduced U373 cells migration (41%) when compared to control group (Figure 11D), suggesting that PLA NPs may have an inhibitory effect on cellular migration in this tumor cell line.

The results for non-tumor cells lines (Figure 11H–J) showed that both nanoparticles were unable to change the migration on FGH, HUVEC and NGM cells.

The results (Figure 12) of the tubulogenesis assay performed on HUVECs cells test demonstrated that when exposed to polylactic acid polymeric nanoparticles or MMSN no effect on tubulogenesis was observed.

The human tumor cells lines used in this work study are under constant investigation by other groups that have contributed to information about the behavior, mechanisms and treatment efficacy [45,58,59,60,61,62,63,64,65,66,67,68,69]. Nanomedicine, emerge as a promising platform for theranostic strategies [65]. For example, glioblastoma therapy faces issues as restricted passage across the blood–brain barrier (BBB) and low enhanced permeability and retention effect. Nanomedicine, emerge as a promising platform for drug delivery to brain malignancies [62]. Also, the current treatment options for prostate cancer are inadequate due to its heterogeneity characteristics. NPs have been developed for disease diagnosis and targeted drug delivery systems [63,64].

In spite of the use of nanotechnology as therapeutics and/or diagnose strategies, there are not sufficient studies about the risk of nanoparticles in their behavior in cellular and molecular level., Once in the blood stream, NPs will act on normal human cells despite its use as theranostic strategies. Healthy human cells used in this study have a systemic distribution. Melanocytes are melanin-producing cells present in skin, eyes, hair, nasal cavity, and inner ear. Molecular alterations in these cells may originate in melanoma [67]. Fibroblasts are the most common type of connective tissue in animals and make up the structural framework in animal tissues. Fibroblasts have crucial role in inflammation, its presence contribute to several pathologies, such as cardiovascular and cancer [68,69]. Endothelial cells play a role as a regulatory interface between blood and tissues, and are a determinant factor in multidrug resistance. Once inside the body, NPs will interact with endothelial cells at some point regardless of their intended destination [20,70,71].

Data showed that an acute exposure to both, polymeric nanoparticles or MMSN, did not show any relevant toxic effects neither tumor cells nor non-tumor cells, corroborating that although nanodrugs may present unrevealed aspects, under acute exposition to human cells are harmless.

## 3. Materials and Methods

### 3.1. Reagents

The following culture reagents were purchased from Sigma–Aldrich (St. Louis, MO, USA): bovine serum albumin (BSA), penicillin, streptomycin, trypsin, Dulbecco’s Modified Eagle Medium (DMEM), and trypan blue. Fetal bovine serum (FBS) was purchased from Cultilab (Campinas, SP, Brazil). The 3-(4,5-dimethylthiazol-2-yl)-2-5-diphenol tetrazolium bromide (MTT) assay was purchased from Sigma–Aldrich. The BCA Protein Assay Kit was purchased from Thermo Scientific (Waltham, MA). For cytochemistry, TRITC-labelled phalloidin was purchased from Sigma–Aldrich and 4,6-diamino-2-phenylindole (DAPI) ProLong1 was obtained from Invitrogen (Carlsbad, CA, USA). For electrophoresis, the PVDF membranes and RainbowTM molecular weight markers were obtained from GE Healthcare (San Francisco, CA, USA), and the enhanced chemiluminescent(ECL) system (Super Signal West Pico chemiluminescent substrate kit) was purchased from Pierce Biotechnology (Rockford, IL, USA). Protease inhibitors phenylmethylsulfonyl fluoride (PMSF) and leupeptin were purchased from Sigma–Aldrich.

### 3.2. Nanoparticles Fabrication and Characterization

#### 3.2.1. MMSN—Small Nanoparticles (50 nm)

The MMSN were produced as describe by Portilho et al. [42]. Briefly, 100 mg of cetrimonium bromide (CTAB)were dissolved in 10 mL of water, followed by addition of 0.74 mL of the ferrofluid (8.88 mg/mL). The mixture was placed in a probe sonicator (Branson 450 Sonifier) for 2 min. The final solution was heated to 65 °C to evaporate the solvent. The resulting suspension was added to a solution of 30 mL of water and 0.548 mL of ammonia (32%), and heated up to 75 °C. Then, 0.5 mL of tetraethyl orthosilicate (TEOS) was added followed by addition of 3 mL of ethyl acetate and stirred at 350 rpm at 75 °C during 3 h. Afterward, the sample was washed with ethanol twice and dried under vacuum overnight. The final magnetic core mesoporous silica was calcined in air at 550 °C for 5 h.

#### 3.2.2. Characterization Techniques

The synthesized materials were characterized by powder X-ray diffraction (PXRD), transmission electron microscopy (TEM) and N_2_ adsorption-desorption analysis. PXRD measurements were obtained using a Bruker AXS D8 Advance diffractometer equipped with CuKα radiation and working at 40 kV/40 mA. PXRD measurements were performed at high-angle (2θ = 15°–68°) and low-angle range (2θ = 1.3°–8.3°). TEM images were taken on a 100 kV JEOL JEM-1010 microscope operated with AMT image capture engine software. TEM samples were prepared by adding 10 µl of nanoparticles suspended in distilled water onto carbon-coated copper grids. The statistical analysis of the data obtained from TEM images was performed using Origin Pro software. N_2_ adsorption-desorption measurements were conducted in a TriStar II Plus surface area and porosity analyzer from Micromeritics. The specific surface area of the material was determined from the adsorption-desorption isotherm by applying the Brunauer-Emmett-Telle (BET) model. The pore volume and average pore size was estimated by using the Barret-Joyner-Halenda (BJH) model.

#### 3.2.3. Polylactic Acid (PLA) Polymeric Nanoparticles 

Polylactic acid (PLA) polymeric nanoparticles were prepared by double emulsion solvent evaporation technique as described by Rosa et al. [58]. Briefly, 2 mL of 1 wt% PVA solution was dripped into 3 mL of dichloromethane, with 100 mg of PLA (with a molar mass of 60,000 g/mol). The mixture sonicated (UP100H, Hielscher, Teltow, Germany) for 1 min at 20% of amplitude to produce a water-in-oil (O/W) emulsion. This emulsion was emulsified again with 60 mL of PVA 1 wt% (pH 11) solution by ultra-turrax (Ultra Turrax^®^ (T10), IKA, Staufen, Germany) processing for 5 min (10,000 rpm) to produce a W/O/W emulsion. The solvents were evaporated under reduced pressure for 1 h at 25 °C (Rotavapor R114, Buchi, Postfach, Switzerland). The nanoparticles were recovered by centrifugation (20,000 rpm for 20 min) Centrifugal Beckman Coulte TM J 25, Brea, California, USA) and washed twice with distilled water to remove the excess of PVA.

#### 3.2.4. Size Determination by Dynamic Light Scattering (DLS)

The size determination as the polydispersity index from the polymeric nanoparticles has been calculated by DLS. The analyze was performed using a Zetasizer Nano ZS (Malvern Instruments, Malvern, UK) under the following conditions: temperature: 25 °C, laser incidence: 173. Both mean and standard deviation (SD) has been accessed.

#### 3.2.5. Raman Analysis

Raman analysis was done on the polymeric nanoparticles using a Horiba–Jobin–Yvon triple spectrometer (model T64000), equipped with a confocal, liquid-N2-cooled and a state solid laser (LAS-532-100-HREV, Kioto, Japan). The wavelength used was 532.0 nm at 14 mW. The samples were irradiated by the laser and the adjustments were done using a microscopy (Olympus, Tokyo, Japan) with objective lens MPLN 100X. The analysis has been done in different points of the surface as varying along the vertical direction. 

### 3.3. Tumor Cell Cultures

MV3 human melanoma cells previously selected by others from a highly metastatic human melanoma fragment [72] were donated by Cezary Marcinkievicz, Temple University, Center for Neurovirology and Cancer Biology (Philadelphia, PA, USA). Human glioma commercial cell lineage (U373 MG), human breast cancer cell lines (MCF-7 and MDA-MB-231), human prostate cancer cell lineage (PC-3), human gastric adenocarcinoma (AGS) and human colorectal adenocarcinoma cell lineage (HT-29) were obtained from Cell Bank of Rio de Janeiro (Rio de Janeiro, Brazil). The MV3, U373 MG, MCF-7, MDA-MB-231, PC-3, AGS and HT-29 cells were routinely maintained in DMEM supplemented with 10% FBS, NaHCO3 (3.7 g/L), 4-(2-hydroxyethyl)-1-piperazineethanesulfonic acid(HEPES) (5.2 g/L), penicillin (0.5 U/mL) and streptomycin (0.5 mg/mL). Cells were incubated at 37 °C in a humidified atmosphere of 5% CO_2_. Cells were grown to confluence culture flasks. Cells were detached by brief treatment with trypsin (0.1%)/ ethylenediaminetetraacetic acid (EDTA) (0.01%).

### 3.4. Non-Tumor Cell Cultures

Human melanocytes (NGM) and human gengival fibroblast (FGH) cell lines were obtained from Cell Bank of Rio de Janeiro, Brazil (0190). NGM and FGH cells were routinely maintained in DMEM and DMEM/F12 medium supplemented with 10% FBS, NaHCO_3_ (3.7 g/L), HEPES (5.2 g/L), penicillin (0.5 U/mL) and streptomycin (0.5 mg/mL), respectively. Cells were incubated at 37 °C in a humidified atmosphere of 5% CO_2_. Cells were grown to confluence in 75 cm² culture flasks and were detached by brief treatment with trypsin (0.1%)/EDTA (0.01%).

Human umbilical vein endothelial cells (HUVECs) were obtained by a modification of the procedure previously described [73], and grown in 199 medium (M199, Sigma Aldrich, St. Loius, MO, USA) supplementedwith 20% fetal calf serum (FCS, Cultilab, Campinas, Brazil) and antibiotics. Endothelial cells were used at passage 3.

### 3.5. In Vitro Design

All nanoparticles developed and characterized in Section 3.2 were evaluated in the following human cells (Section 3.3 and Section 3.4), divided in tumor and non- tumor cells.

### 3.6. Flow Cytometry Analysis

Cells (2 × 10^5^ cells) were seeded in 24-well plates and allowed to attach for 24 h. Then, cells were treated in presence or absence of nanoparticles (20 µg/mL) for another 24 h. Fluorescence-activated cell sorting (FACS) analysis was conducted with an ACCURI C6 flow cytometer with CFLOW software (Becton Dickinson, Heidelberg, Germany). Following incubation, the cells were detached and centrifuged. Then, pellets were re-suspended in FACs Binding Buffer for forward scatter/side scatter (FSC/SSC) light scattering analysis.

### 3.7. Cell Cycle Analysis

Cells (2 × 10^5^ cells) were seeded in 24-well plates and allowed to attach for 24 h. Then, cells were treated in presence or absence of nanoparticles (20 µg/mL) for another 24 h. Then cells were harvested, fixed in 70% ethanol at −20 °C, and were then stained with propidium iodide (PI; 50 µg/mL) containing RNase A (30 µg/mL) (both from Sigma) at 37 °C for 30 min. FACS analysis was conducted with an ACCURI C6 flow cytometer with CFLOW software (Becton Dickinson, Heidelberg, Germany).

### 3.8. Cytochemistry Assay

Tumor cells were seeded in 24-well plates on sterile glass coverslips and allowed to attach for 24 h. Then, cells were treated in the presence or absence of nanoparticles (20 µg/mL) for 24 h and then the cells from both group (treated and untreated) were washed and fixed with a solution of PFA (paraformaldehyde 4%), sucrose and phosphate-buffer-saline (PBS) for 20 min at room temperature. Then the cells were permeabilized using a solution made of Triton X-100 (4%) and PBS for 5 min. In the final step, cells were labeled with TRITC-phalloidin (1:1000) for 2 h at room temperature. In the end, the cells were mounted for nuclear staining using a Prolong Gold antifade reagent with 4,6-diamino-2-phenylindole (DAPI). The cells from both group were evaluated using a Olympus B×40 microscopy prepared for epifluorescence.

### 3.9. Proliferation Assay

Tumor cells (5 × 10^3^ cells/well) were seeded and allowed to attach for 24 h. Then, cells were treated in the presence or absence of nanoparticles (20 µg/mL) for another 24 h. After washing, the number of attached cells was determined using the MTT assay as described by [74].

### 3.10. Wound-Healing Assay

Cells (2 × 10^5^ cells) were seeded in 24-well plates and allowed to attach for 24 h. Wounds were created in confluent cells using a pipette tip. Cells were then rinsed with medium to remove floating cells and debris. Before treatment, cells were incubated or not with Mitomicin C (5 µg/mL) for 1 h. After washing, cells were treated in the presence or absence of nanoparticles (20 µg/mL) for another 24 h. The culture plates were incubated at 37 °C. Wounds were measured at 0 and 24 h. The percentage of wound-healing was calculated using the equation: (percentage wound-healing) = average of ([gap length: 0 h]-[gap length: 24 h])/[gap length: 0 h]).

### 3.11. Immunoblotting

Cells (2 × 10^5^ cells) were seeded in 24-well plates and allowed to attach for 24 h. Then, cells were treated in the presence or absence of nanoparticles (20 µg/mL) for another 24 h. The total protein were performed using the bicinchoninic acid assay (BCA). Also, were analyzed with the following primary antibodies: anti-p-ERK (1:1000), anti-ERK (1:1000), anti-p-Akt (1:1000) and anti- Akt (1:1000) were from Cell Signaling; anti-α-tubulin (1:1000) was from Sigma, anti-GAPDH (1:1000) was from Ambion The secondary antibodies against rabbit IgG and mouse IgG from Dako (1:2000) were from Cell Signaling.

### 3.12. In Vitro Tubulogenesis Assay

HUVECs (5 × 10^4^ cells) were seeded in triplicate wells onto Matrigel (growth factor reduced, BD Biosciences, São Paulo, Brazil), in M199 medium supplemented with 5% FCS in the presence or absence of nanoparticles (20 µg/mL). After 16 h, cells were fixed in 1.1% glutaraldehyde in PBS for 10 min and then were washed with PBS. Tube-like structures/well was counted by two independent observers.

### 3.13. Statistical Analysis

The statistical analyses for multiple comparisons were performed using one-way analysis of variance (ANOVA) followed by Bonferroni post-test analysis. For comparisons between two groups, we used an unpaired *t*-test. We considered statistically significant differences between groups with *p* < 0.05. Analyses were made using the GraphPad Prism software for Windows.

## 4. Conclusions

All the data presented here demonstrate that acute exposure to polylactic acid polymeric nanoparticles or MMSN did not have relevant toxic effects either on non-tumor or tumor cells, representing an advantage for the use of both nanoparticles in nanomedicine as nanodrugs.

## Figures and Tables

**Figure 1 ijms-21-00230-f001:**
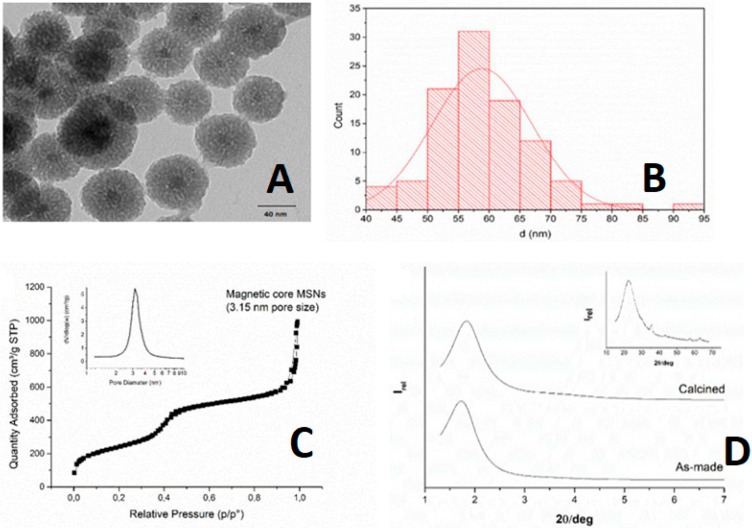
(**A**) Transmission electron microscopy showing the size as the magnetic core of the mesoporous silica nanoparticles. (**B**) Size histogram and normal size distribution of magnetic core mesoporous silica. (**C**) N_2_ adsorption-desorption isotherm of magnetic mesoporous silica nanoparticles (MMSN), showing the pore size. (**D**) Powder X-ray diffraction patterns of (bottom) as-made magnetic core MSNs (**S0-1**) and calcined magnetic core MSNs (**S0-2**).

**Figure 2 ijms-21-00230-f002:**
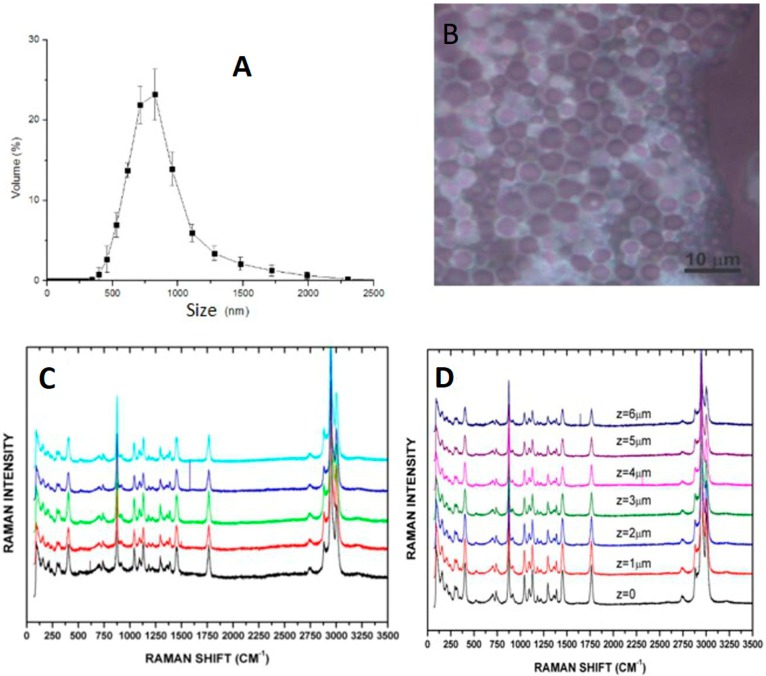
(**A**) Dynamic light scattering (DLS) size distribution of giant polymeric nanoparticle (GPPM). (**B**) Raman analysis corroborating the monomodal behavior. (**C**) Raman analysis showing the system overview and varying in axis y and Z (**D**) corroborating the uniformity of the microparticles tested and the emptiness state of the nanoparticle system.

**Figure 3 ijms-21-00230-f003:**
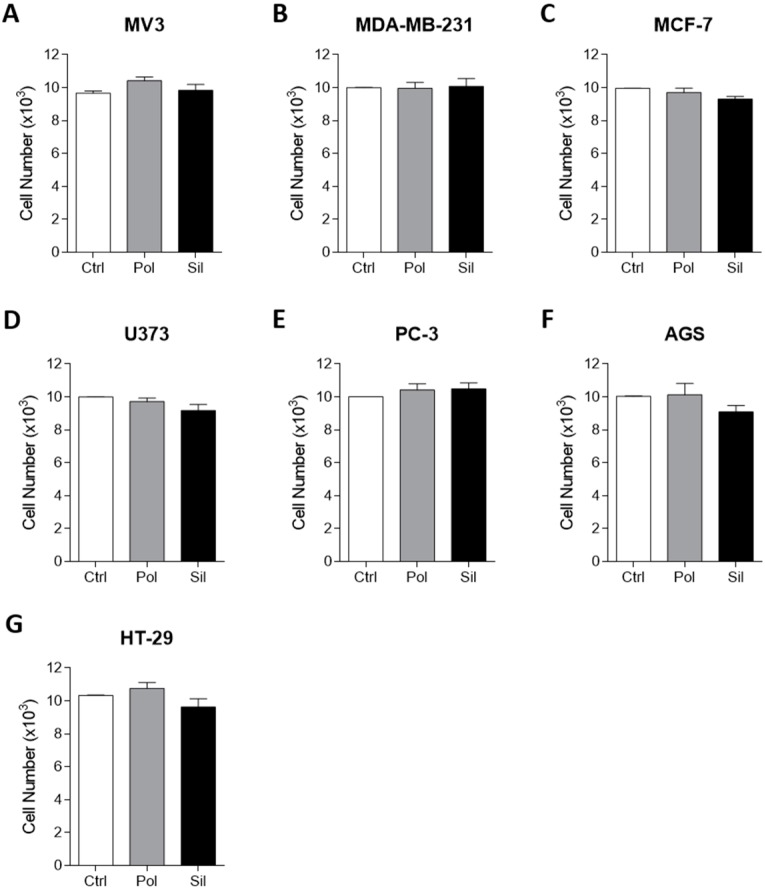
Nanoparticle effects on tumor cytotoxicity. Tumor cells were incubated with polymeric or silica nanoparticles (20μg/mL) for 24 hs. Cytotoxicity was evaluated using the 3-(4,5-dimethylthiazol-2-yl)-2,5-diphenyltetrazolium bromide MTT assay. (**A**). MV3 (human melanoma cancer cell line) (**B**). MDA-MB-231 (human triple negative breast cancer cell line) (**C**). MCF-7 (human breast cancer cell line) (**D**). U373 (human glioblastoma cell line) (**E**). PC-3 (human prostate cancer cell line) (**F**). AGS (human gastric cancer cell line) (**G**). HT-29 (human colon cancer cell line). Results are presented as the mean ± SD calculated from three individual experiments (* *p* < 0.05).

**Figure 4 ijms-21-00230-f004:**
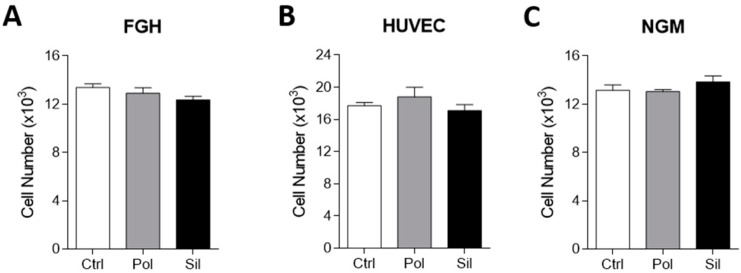
Nanoparticles effects on non-tumor cytotoxicity. FGH (human fibroblast cell line), HUVEC (Human umbilical vein endothelial cell line) and NGM (human melanocyte cell line derived from blue nevus cell line cells were incubated with polymeric (pol) or silica (sil) nanoparticles (20 µg/mL) for 24 hs. Cytotoxicity was evaluated using the MTT assay. (**A**). FGH (**B**). HUVEC (**C**). NGM. Results are presented as the mean ± SD calculated from three individual experiments (* *p* < 0.05).

**Figure 5 ijms-21-00230-f005:**
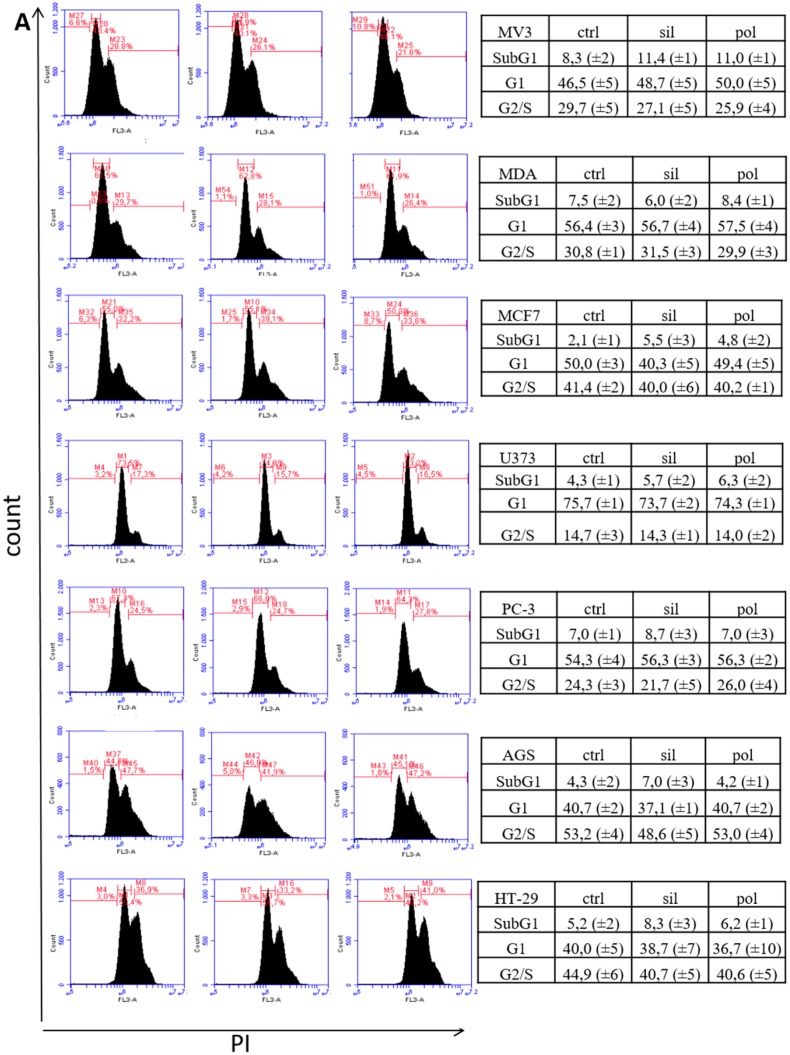
Effects of nanoparticles in the cell cycle of tumor and non-tumor cells. Tumor cells (panel **A**) and non-tumor cells (panel **B**) were incubated with polymeric (pol) or silica (sil) nanoparticles (20 μg/mL) for 24 h. Then, cells were harvest and dyed with Propidium iodide (PI) as described in methodology section. DNA content was detected by flow cytometry for the presence of cells in different stages (SubG1, G1 and S/G2) of cell cycle. Figures are representative from 3 individual experiments.

**Figure 6 ijms-21-00230-f006:**
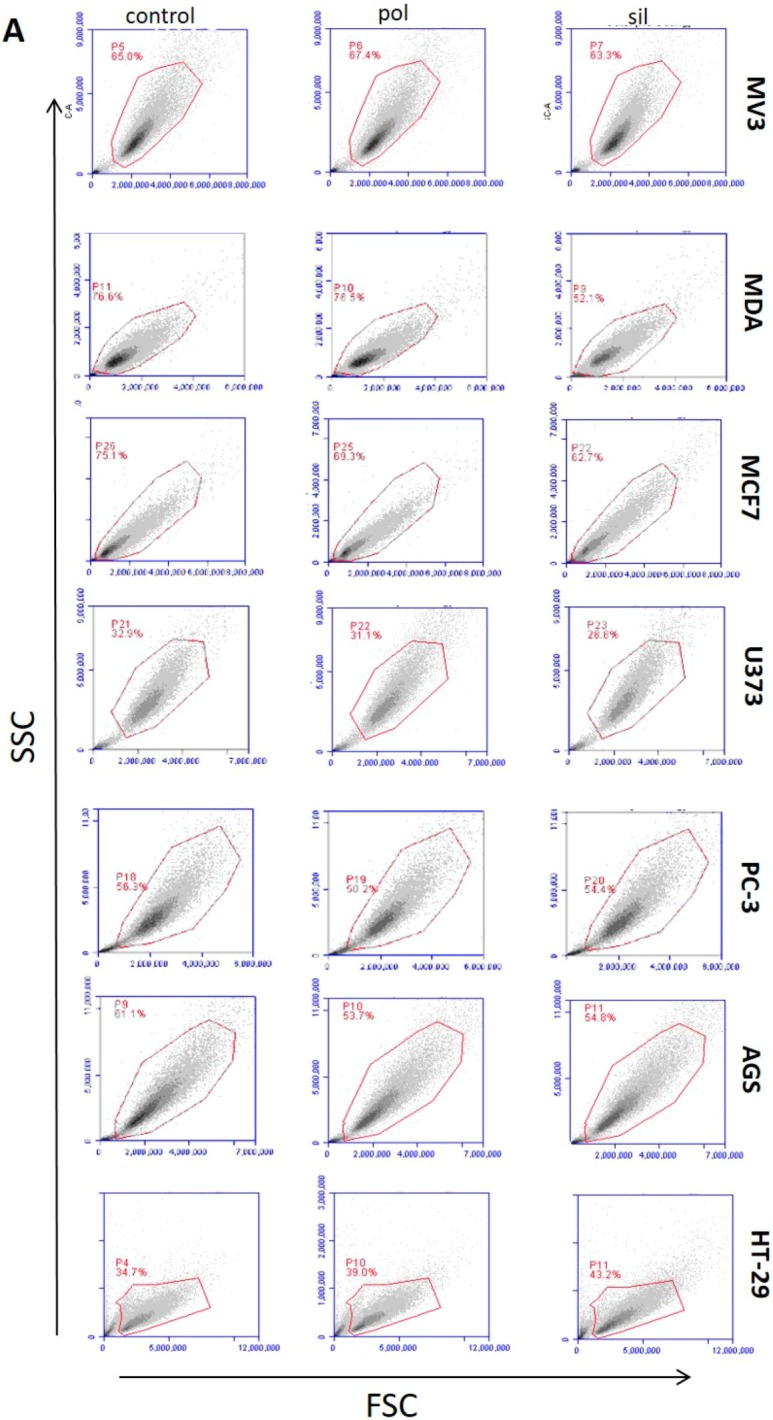
Effects of nanoparticles exposure in tumor. (**A**) MV3, MDA-MB-231, MCF-7, U373, PC-3, AGS, HT-29 and non-tumor (**B**) FGH, HUVEC, NGM cells.

**Figure 7 ijms-21-00230-f007:**
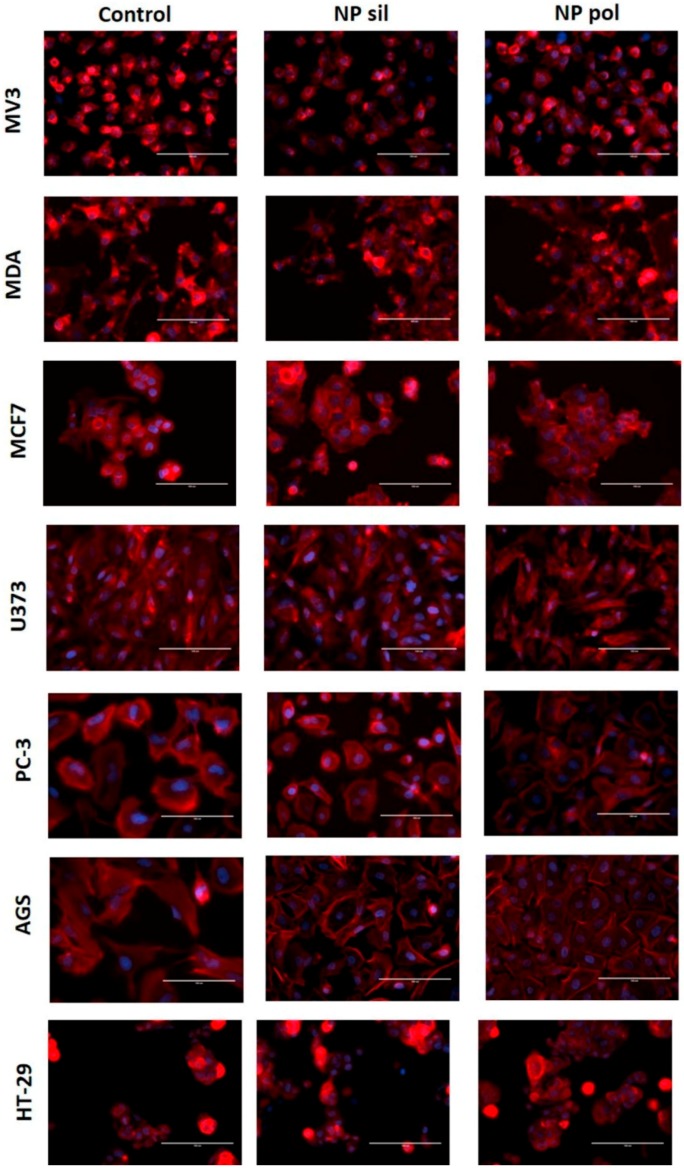
Effects of nanoparticles in tumor cells actin cytoskeleton. Tumor cells were incubated with polymeric (pol) and silica (sil) nanoparticles (20 µg/mL) for 24 h. Filamentous actin was labeled with tetramethylrhodamine (TRITC)-phalloidin and nuclei were stained with 4,6-diamino-2-phenylindole (DAPI). Representative images from three individual experiments were captured at 400× magnification.

**Figure 8 ijms-21-00230-f008:**
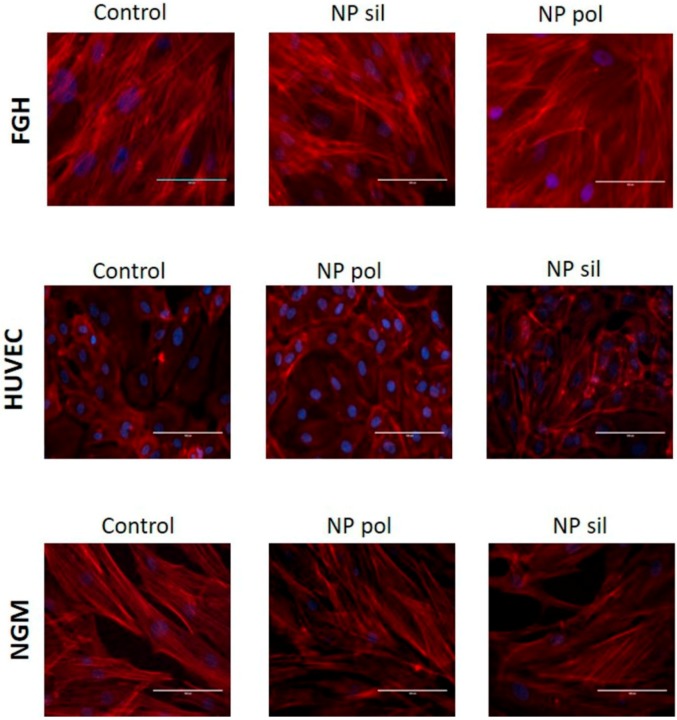
Effects of nanoparticles in non-tumor cells actin cytoskeleton. FGH, HUVEC and NGM cells were incubated with polymeric (pol) or silica (sil) nanoparticles (20 µg/mL) for 24 h. Filamentous actin was labeled with TRITC-phalloidin and nuclei were stained with DAPI. Representative images from 3 individual experiments were captured at 400× magnification.

**Figure 9 ijms-21-00230-f009:**
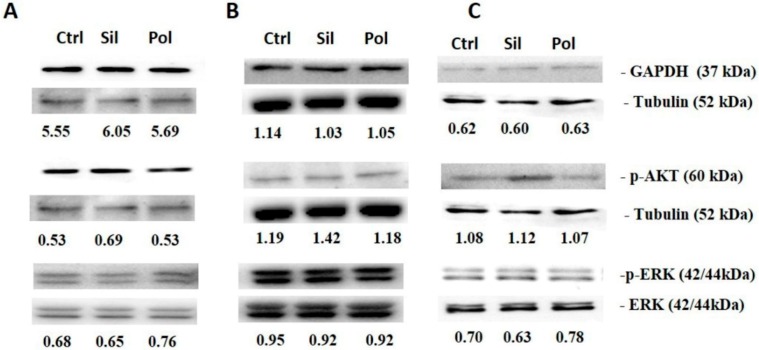
Nanoparticles effects on non-tumor intracellular signaling. FGH, HUVEC and NGM cells were incubated with polymeric (pol) or silica(sil) nanoparticles (20 μg/mL) for 24 h. Then, cell lysates were immunoblotted with anti-protein kinase R (PKR)-like endoplasmic reticulum kinase (PERK)–alpha 1 kinase 2(anti-pERK1/2) and anti-extracellular signal-regulated Kinase 1/2 (ERK1/2), anti-phospo-protein kinase B(pAKT) ser 473 and anti- protein kinase-B (AKT), anti- Glyceraldehyde 3-phosphate dehydrogenase(GAPDH) and anti-tubulin. (**A**). FGH (**B**). HUVEC (**C**). NGM. Results are presented as the mean ± SD calculated from three independent experiments (* *p* < 0.05).

**Figure 10 ijms-21-00230-f010:**
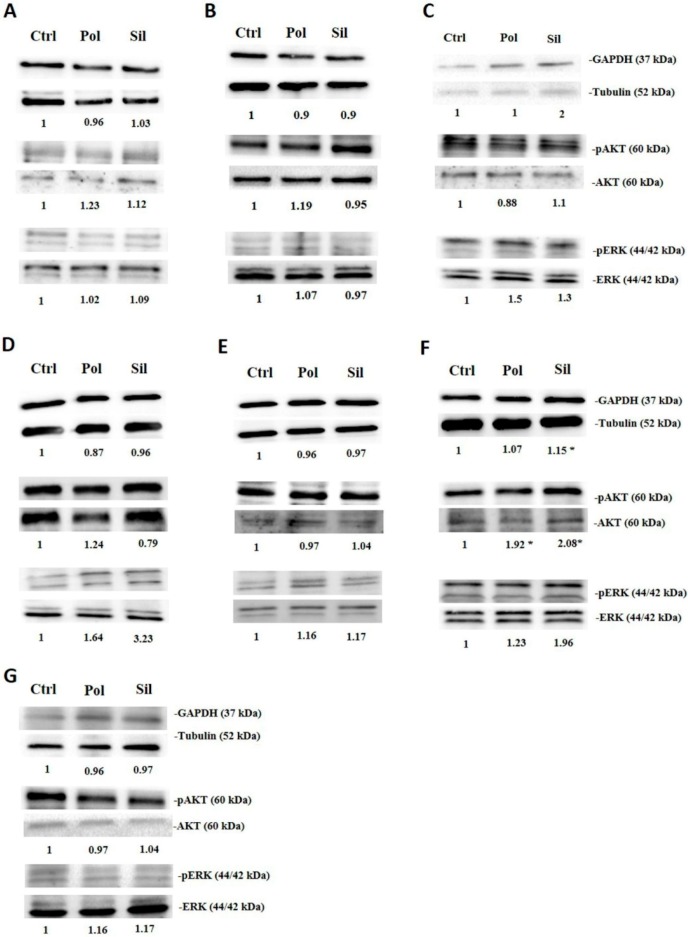
Nanoparticles effects on tumor intracellular signalling. Tumor cells were incubated with polymeric (pol) or silica (sil) nanoparticles (20 μg/mL) for 24 h. Then, cell lysates were immunoblotted with anti-pERK1/2 and anti-ERK1/2, anti-pAKT ser 473 and anti-AKT, anti-GAPDH and anti-tubulin. (**A**). MV3 (**B**). MDA-MB-231 (**C**). MCF-7 (**D**). U373 (**E**). PC-3 (**F**). AGS (**G**). HT-29. Results are presented as the mean ±SD calculated from three independent experiments (* *p* < 0.05).

**Figure 11 ijms-21-00230-f011:**
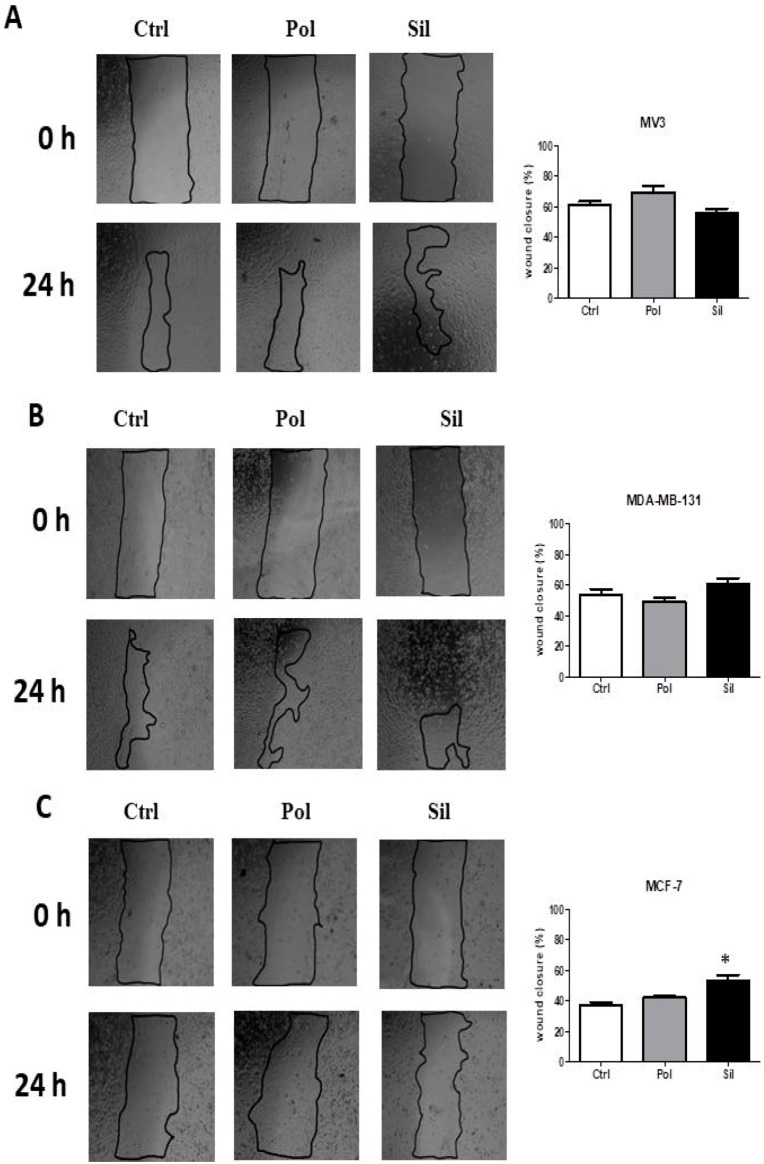
Effects of nanoparticles in tumor and non-tumor cells migration. Tumor cells ((**A**). MV3 (**B**). MDA-MB-231 (**C**). MCF-7 (**D**). U373 (**E**). PC-3 (**F**). AGS (**G**). HT-29) and non-tumor ((**H**). FGH, (**I**). HUVEC, (**J**). NGM) cells were cultured for 24 h in the absence (ctrl) or in the presence of 20 ug/mL of polymeric (Pol) or silica (Sil) nanoparticle. Mean e standard variation were calculated from 3 individual experiments (* *p* < 0.05).

**Figure 12 ijms-21-00230-f012:**
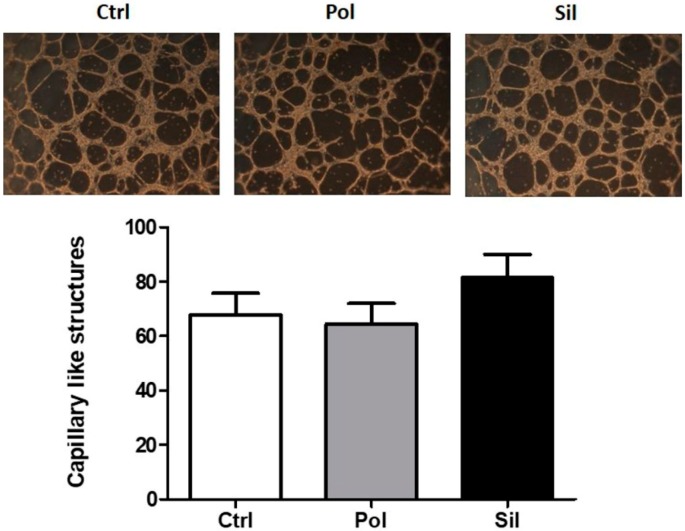
Effects of nanoparticles in cell migration and tubulogenesis HUVEC were cultured for 24 h in the absence (ctrl) or in the presence of 20 ug/mL of polymeric (Pol) or silica (Sil) nanoparticle Endothelial sprouts were quantified by counting the sprouts in four high-power microscopic fields (at 100× magnification) in each treatment. (**A**) Representative structures formed; (**B**) total number of structures formed by cells. Results are shown as the mean obtained from three individual experiments (* *p* < 0.05).

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
