# Peer review of "Molecular and Cellular Risk Assessment of Healthy Human Cells and Cancer Human Cells Exposed to Nanoparticles"

_ijms, 2019, doi:10.3390/ijms21010230_

Round 1
Reviewer 1 Report
The current manuscript is really novel and the authors have performed extensive experimental studies to support their claim. Overall, the article is very well presented and written. Hence, I am recommending to accept this paper as it's present form.
Author Response
Response Letter
Manuscript: Molecular and Cellular Risk Assessment of Primary Human Cells and Cancer Human Cells Exposed to Nanoparticles- ijms-622197
Dear Dr. Zurini Fu
Assistant Editor
International Journal of Molecular Science
Please find below a point-by-point response to comments made by the Referees on our manuscript.
Sincerely,
Ralph Santos-Oliveira
RESPONSES TO THE COMMENTS
Reviewer #1
The present paper by Edward Helal-Neto, propose the molecular and cellular risk assessment of primary human cells and cancer human cells acute exposed to nanoparticles. Even with good experimental methods there are some major points that should be addressed.
First the use of ‘primary human cells’ in the tittle is a little misleading, in the manuscript not refers as primary human cells.Answer. We thank the comment made by the Reviewer. As suggested, we modified the title in the revised manuscript version and changed to healthy instead of primary.
In the introduction, in my opinion reference 29 its not related with mesoporous silica nanoparticles.Answer. We're sorry for the mistake and grateful for correction. Modifications were made in the revised version.
Second, I find the results and discussion presentation a lack of depth and for instances:‘2.1.1. MMSN’ taking in consideration the figure (A,B,C) the presented of results should respect the right order.
‘2.1.2. Polylactic acid (PLA) polymeric nanoparticle’ presents a figure for Raman analysis and in the results and discussion of manuscript not indicates clearly that correlation.
Answer. We thank the comment made by the Reviewer. As suggested, we modified the text in the revised manuscript version. We have re-done all the figures as we also have inserted some explanation in the text.
3 ‘2.3. Cell morphology’ Figure 7 and Figure 8 should appear in the right order. ‘2.5. Fig.12 E, doesn’t exist.
Answer. We're sorry for the mistake and grateful for correction. Modifications were made in the revised version.
5 In the materials and methods, in my opinion the table could be removed, and the methodology better explained.
Answer. We thank the comment made by the Reviewer. As suggested, we removed the table and explained more carefully the methodology section in the revised manuscript version.
In general overview, the manuscripts as major points to be reconsidered and modify:
6 the order of the reference’s appearance in the manuscript, correct the letter size, some spaces needed to be corrected, consistence in Figure/Fig. and in the order that appears in the manuscript, List of references must be revised and add the acknowledgements and author contributions.
Answer. We're sorry for the mistake and grateful for correction. Modifications were made in the text of the revised version.
Reviewer #2
Helal-Neto et al. have investigated the effect of PLA nanoparticles and MMSN on cellular toxicity in vitro. Strength of this study is the number of cell lines tested and the extensive repertoire of different assays linked to cell function and survival. However, I have some points which should be addressed:
NP have been shown to increase the production of reactive oxygen species in cells, and this is a very sensitive marker of cell activation, which also plays a role in cell survival. This should also be measured in the different cell lines incubated with the different nanoparticles.
Answer. We thank and agree with the comment made by the Reviewer. Unfortunately, the Brazilian government has cut more than half of the financial budget for science and technology. Thus, although we agree with the importance of ROS as a marker of cellular activation, we are unable to acquire the necessary reagents for this measurement now. However, we believe that the data provided by other assays, such as actin labeling and blotting, also provide relevant information on the activation of the cells studied. Changes in the dynamics of the actin microfilaments may result on different cell destiny. A loss of actin polymerization could lead to cell detachment which, in ultimately case, may result in cell death. However, actin microfilaments rearrangement can be associated with the activation of signaling pathways that regulates cell behavior [Papakonstanti, E.A. & Stournaras, C. Cell responses regulated by early reorganization of actin cytoskeleton. FEBS Lett. 2008 18;582(14), 2120-7]. Also, changes in actin polymerization can be a consequence of cell migration when it is observed an increase in the formation of stress fiber in the migrating cell [Hanna, S. & El-Sibai, M. Signaling networks of Rho GTPases in cell motility. Cell Signal 2013.25, 1955-6]. We believe that in a near future a more complete analysis will be done in our group, however at this time we do request a understanding from the reviewer at this regard.
2. The authors use 20mg/ml NP in their experiments. Is this equivalent to the concentration, which would be used in vivo?
Answer. We thank the comment made by the Reviewer. Modifications were made in the text explained the use of this dose in the revised version. The chosen dose in this work has been used by our group in several other studies correlating the in vitro and in vivo. Since we consider a acute exposition.
- Pinto, S.R. et al. Cytotoxicity, genotoxicity, transplacental transfer and tissue disposition in pregnant rats mediated by nanoparticles: the case of magnetic core mesoporous silica nanoparticles. Artif Cells NanomedBiotechnol. 2018 24, 1-12.
- Portilho, F.L. et al. Magnetic core mesoporous silica nanoparticles doped with dacarbazine and labelled with 99mTc for early and differential detection of metastatic melanoma by single photon emission computed tomography.Artif Cells NanomedBiotechnol. 2018 27, 1-8
- de Jesus Felismino, C. et al. Effect of obesity on biodistribution of nanoparticles. .J Control Release. 2018 281, 11-18
Have the authors tested a higher concentration and observed a cut-off point, where NP do cause cytotoxicity?
Answer. We thank the comment made by the Reviewer. However, because of the low budget, we had to choose just one dose. In this study we worked with 10 different cell lines and we do not have financial support to test different concentrations. So, as answered above, we opted to choose a dose that is already used in vivo assays. Despite of this, we have another study where we are testing MMSN and PLA NPs in different concentrations (20 and 50 uM) using MDA-MB-231 cells. Preliminary data demonstrated that after 24 hours and 48 hours there were no differences in the cell viability in both groups.
Please add quantification analyses of the filamentous actin labelling studies in Figure 7.
Answer. We thank the comment made by the Reviewer. As suggested, we added a graphics that shows the quantification analysis of the filamentous actin labelling as a supplementary figure in the revised manuscript version.
Figure S01. Effects of nanoparticles in non-tumor cells actin cytoskeleton. Human cells were incubated with polymeric (pol) or silica (sil) nanoparticles (20µg/mL) for 24hs. Filamentous actin was labeled with TRITC-phalloidin and nuclei were stained with DAPI. Representative graphics showing the relative fluorescence intensity of TRITC-phalloidin from 3 individual experiments captured at 400x magnification *p<0.05.
Supplementary methodology: fluorescence quantification analyses
Cells (7 x 104 cells/well) were cultured on sterile glass coverslipsthen for 24 hours. Then, cells were fixed with 4% paraformaldehyde in sucrose/PBS solution for 20 min at room temperature, permeabilized with Triton X-100 (0.1%)/PBS for 5 min and blocked with 5% BSA for 30 minutes. Then, cells were labeled with TRITC-phalloidin (1:400) overnight at 4°C. Finally, slides were mounted using ProLong Gold antifade reagent with 4,6-diamidino-2-phenylindole (DAPI) for nuclear staining. Coverslips were examined under an Olympus BX40 microscope equipped for epifluorescence at 400x magnification. Fluorescence intensity was measured on Icy bioimage analysis version 1.9.5.1 and Excel. The result shows the mean ± SD of the relative fluorescence intensity.
Figure 11C and F: The photos, which should be representative, do not fit to the results in the respective graph.
Answer. We're sorry for the mistake and grateful for correction. During the review of the figures we noticed that we made a mistake in analyzing the figures. In MCF-7 and AGS cells, the values ​​of pol and sil groups were switched. All analyzes were redone. We also observed the same error in PC3 cell migration. The new version has the correct graphic bars as well as their corrections in the text. As you can read below
2.5 Cell migration and Tubulogenesis
…“However, polylactic acid polymeric nanoparticles increased the wound closure of PC3 (39%) cells (figure 11 E) when compared to untreated group (controls). It suggests that PLA NPs may induce cellular migration in this tumor cell line. Similarly, when MCF-7 and AGS cells were treated with Magnetic Core Mesoporous Silica nanoparticles, it increased the cell migration (58% and 53%, respectively) when compared to control group (Fig. 11 C and F)”…
Materials and Methods: Please add the source of the used antibodies.
Answer. We thank the comment made by the Reviewer. Modifications were made in the revised version.
Minor points:
The English needs to be corrected by a native speaker. There are several spelling errors (citotoxicity instead of cytotoxicity etc.), grammar mistakes (“we did not observed” instead of “we did not observe”, the word “it” as in “it is” is missing several times etc.) and general wording (“To have a better understand..” instead of “To become a better understanding…” etc.).
In the section 2.4 Intracellular signalling the authors write on several occasions: “the expression of Akt and ERK1/2 phosphorylated”. The phosphorylation of proteins is not expressed, phosphorylation is a posttranslational modification. Please rephrase.
Figure legend to figure 4: It states Tumor cells were incubated…. This is not correct, as FGH, HUVEC and NGM cells were used. It is probably a copy-paste mistake from Figure 3. Please correct.
Answer: The full text has been corrected by a native speaker (Dr. Frank Alexis). We are sorry for the previous mistakes.

Reviewer 2 Report
The present paper by Edward Helal-Neto, propose the molecular and cellular risk assessment of primary human cells and cancer human cells acute exposed to nanoparticles. Even with good experimental methods there are some major points that should be addressed.
First the use of ‘primary human cells’ in the tittle is a little misleading, in the manuscript not refers as primary human cells. In the introduction, in my opinion reference 29 its not related with mesoporous silica nanoparticles.
Second, I find the results and discussion presentation a lack of depth and for instances:
‘2.1.1. MMSN’ taking in consideration the figure (A,B,C) the presented of results should respect the right order.
‘2.1.2. Polylactic acid (PLA) polymeric nanoparticle’ presents a figure for Raman analysis and in the results and discussion of manuscript not indicates clearly that correlation.
‘2.3. Cell morphology’ Figure 7 and Figure 8 should appear in the right order.
‘2.5. Fig.12 E, doesn’t exist.
In the materials and methods, in my opinion the table could be removed, and the methodology better explained.
In general overview, the manuscripts as major points to be reconsidered and modify:
the order of the reference’s appearance in the manuscript, correct the letter size, some spaces needed to be corrected, consistence in Figure/Fig. and in the order that appears in the manuscript, List of references must be revised and add the acknowledgements and author contributions.
Author Response
Response Letter
Manuscript: Molecular and Cellular Risk Assessment of Primary Human Cells and Cancer Human Cells Exposed to Nanoparticles- ijms-622197
Dear Dr. Zurini Fu
Assistant Editor
International Journal of Molecular Science
Please find below a point-by-point response to comments made by the Referees on our manuscript.
Sincerely,
Ralph Santos-Oliveira
RESPONSES TO THE COMMENTS
Reviewer #2
The present paper by Edward Helal-Neto, propose the molecular and cellular risk assessment of primary human cells and cancer human cells acute exposed to nanoparticles. Even with good experimental methods there are some major points that should be addressed.
First the use of ‘primary human cells’ in the tittle is a little misleading, in the manuscript not refers as primary human cells.
Answer. We thank the comment made by the Reviewer. As suggested, we modified the title in the revised manuscript version and changed to healthy instead of primary.
In the introduction, in my opinion reference 29 its not related with mesoporous silica nanoparticles.
Answer. We're sorry for the mistake and grateful for correction. Modifications were made in the revised version.
Second, I find the results and discussion presentation a lack of depth and for instances:
‘2.1.1. MMSN’ taking in consideration the figure (A,B,C) the presented of results should respect the right order.
‘2.1.2. Polylactic acid (PLA) polymeric nanoparticle’ presents a figure for Raman analysis and in the results and discussion of manuscript not indicates clearly that correlation.
Answer. We thank the comment made by the Reviewer. As suggested, we modified the text in the revised manuscript version. We have re-done all the figures as we also have inserted some explanation in the text.
3 ‘2.3. Cell morphology’ Figure 7 and Figure 8 should appear in the right order. ‘2.5. Fig.12 E, doesn’t exist.
Answer. We're sorry for the mistake and grateful for correction. Modifications were made in the revised version.
5 In the materials and methods, in my opinion the table could be removed, and the methodology better explained.
Answer. We thank the comment made by the Reviewer. As suggested, we removed the table and explained more carefully the methodology section in the revised manuscript version.
In general overview, the manuscripts as major points to be reconsidered and modify:
6 the order of the reference’s appearance in the manuscript, correct the letter size, some spaces needed to be corrected, consistence in Figure/Fig. and in the order that appears in the manuscript, List of references must be revised and add the acknowledgements and author contributions.
Answer. We're sorry for the mistake and grateful for correction. Modifications were made in the text of the revised version.
Reviewer 3 Report
Helal-Neto et al. have investigated the effect of PLA nanoparticles and MMSN on cellular toxicity in vitro. A strength of this study is the number of cell lines tested and the extensive repertoire of different assays linked to cell function and survival. However, I have some points which should be addressed:
NP have been shown to increase the production of reactive oxygen species in cells, and this is a very sensitive marker of cell activation, which also plays a role in cell survival. This should also be measured in the different cell lines incubated with the different nanoparticles.The authors use 20mg/ml NP in their experiments. Is this equivalent to the concentration, which would be used in vivo? Have the authors tested a higher concentration and observed a cut-off point, where NP do cause cytotoxicity?
Please add quantification analyses of the filamentous actin labelling studies in Figure 7.
Figure 11C and F: The photos, which should be representative, do not fit to the results in the respective graph.
Materials and Methods: Please add the source of the used antibodies.
Minor points:
The English needs to be corrected by a native speaker. There are several spelling errors (citotoxicity instead of cytotoxicity etc.), grammar mistakes (“we did not observed” instead of “we did not observe”, the word “it” as in “it is” is missing several times etc.) and general wording (“To have a better understand..” instead of “To become a better understanding…” etc.).In the section 2.4 Intracellular signalling the authors write on several occasions: “the expression of Akt and ERK1/2 phosphorylated”. The phosphorylation of proteins is not expressed, phosphorylation is a posttranslational modification. Please rephrase.
Figure legend to figure 4: It states Tumor cells were incubated…. This is not correct, as FGH, HUVEC and NGM cells were used. It is probably a copy-paste mistake from Figure 3. Please correct.
Author Response
Response Letter
Manuscript: Molecular and Cellular Risk Assessment of Primary Human Cells and Cancer Human Cells Exposed to Nanoparticles- ijms-622197
Dear Dr. Zurini Fu
Assistant Editor
International Journal of Molecular Science
Please find below a point-by-point response to comments made by the Referees on our manuscript.
Sincerely,
Ralph Santos-Oliveira
RESPONSES TO THE COMMENTS
Reviewer #3
Helal-Neto et al. have investigated the effect of PLA nanoparticles and MMSN on cellular toxicity in vitro. Strength of this study is the number of cell lines tested and the extensive repertoire of different assays linked to cell function and survival. However, I have some points which should be addressed:
NP have been shown to increase the production of reactive oxygen species in cells, and this is a very sensitive marker of cell activation, which also plays a role in cell survival. This should also be measured in the different cell lines incubated with the different nanoparticles.
Answer. We thank and agree with the comment made by the Reviewer. Unfortunately, the Brazilian government has cut more than half of the financial budget for science and technology. Thus, although we agree with the importance of ROS as a marker of cellular activation, we are unable to acquire the necessary reagents for this measurement now. However, we believe that the data provided by other assays, such as actin labeling and blotting, also provide relevant information on the activation of the cells studied. Changes in the dynamics of the actin microfilaments may result on different cell destiny. A loss of actin polymerization could lead to cell detachment which, in ultimately case, may result in cell death. However, actin microfilaments rearrangement can be associated with the activation of signaling pathways that regulates cell behavior [Papakonstanti, E.A. & Stournaras, C. Cell responses regulated by early reorganization of actin cytoskeleton. FEBS Lett. 2008 18;582(14), 2120-7]. Also, changes in actin polymerization can be a consequence of cell migration when it is observed an increase in the formation of stress fiber in the migrating cell [Hanna, S. & El-Sibai, M. Signaling networks of Rho GTPases in cell motility. Cell Signal 2013.25, 1955-6]. We believe that in a near future a more complete analysis will be done in our group, however at this time we do request a understanding from the reviewer at this regard.
2. The authors use 20mg/ml NP in their experiments. Is this equivalent to the concentration, which would be used in vivo?
Answer. We thank the comment made by the Reviewer. Modifications were made in the text explained the use of this dose in the revised version. The chosen dose in this work has been used by our group in several other studies correlating the in vitro and in vivo. Since we consider a acute exposition.
- Pinto, S.R. et al. Cytotoxicity, genotoxicity, transplacental transfer and tissue disposition in pregnant rats mediated by nanoparticles: the case of magnetic core mesoporous silica nanoparticles. Artif Cells NanomedBiotechnol. 2018 24, 1-12.
- Portilho, F.L. et al. Magnetic core mesoporous silica nanoparticles doped with dacarbazine and labelled with 99mTc for early and differential detection of metastatic melanoma by single photon emission computed tomography.Artif Cells NanomedBiotechnol. 2018 27, 1-8
- de Jesus Felismino, C. et al. Effect of obesity on biodistribution of nanoparticles. .J Control Release. 2018 281, 11-18
Have the authors tested a higher concentration and observed a cut-off point, where NP do cause cytotoxicity?
Answer. We thank the comment made by the Reviewer. However, because of the low budget, we had to choose just one dose. In this study we worked with 10 different cell lines and we do not have financial support to test different concentrations. So, as answered above, we opted to choose a dose that is already used in vivo assays. Despite of this, we have another study where we are testing MMSN and PLA NPs in different concentrations (20 and 50 uM) using MDA-MB-231 cells. Preliminary data demonstrated that after 24 hours and 48 hours there were no differences in the cell viability in both groups.
Please add quantification analyses of the filamentous actin labelling studies in Figure 7.
Answer. We thank the comment made by the Reviewer. As suggested, we added a graphics that shows the quantification analysis of the filamentous actin labelling as a supplementary figure in the revised manuscript version.
Figure S01. Effects of nanoparticles in non-tumor cells actin cytoskeleton. Human cells were incubated with polymeric (pol) or silica (sil) nanoparticles (20µg/mL) for 24hs. Filamentous actin was labeled with TRITC-phalloidin and nuclei were stained with DAPI. Representative graphics showing the relative fluorescence intensity of TRITC-phalloidin from 3 individual experiments captured at 400x magnification *p<0.05.
Supplementary methodology: fluorescence quantification analyses
Cells (7 x 104 cells/well) were cultured on sterile glass coverslipsthen for 24 hours. Then, cells were fixed with 4% paraformaldehyde in sucrose/PBS solution for 20 min at room temperature, permeabilized with Triton X-100 (0.1%)/PBS for 5 min and blocked with 5% BSA for 30 minutes. Then, cells were labeled with TRITC-phalloidin (1:400) overnight at 4°C. Finally, slides were mounted using ProLong Gold antifade reagent with 4,6-diamidino-2-phenylindole (DAPI) for nuclear staining. Coverslips were examined under an Olympus BX40 microscope equipped for epifluorescence at 400x magnification. Fluorescence intensity was measured on Icy bioimage analysis version 1.9.5.1 and Excel. The result shows the mean ± SD of the relative fluorescence intensity.
Figure 11C and F: The photos, which should be representative, do not fit to the results in the respective graph.
Answer. We're sorry for the mistake and grateful for correction. During the review of the figures we noticed that we made a mistake in analyzing the figures. In MCF-7 and AGS cells, the values ​​of pol and sil groups were switched. All analyzes were redone. We also observed the same error in PC3 cell migration. The new version has the correct graphic bars as well as their corrections in the text. As you can read below
2.5 Cell migration and Tubulogenesis
…“However, polylactic acid polymeric nanoparticles increased the wound closure of PC3 (39%) cells (figure 11 E) when compared to untreated group (controls). It suggests that PLA NPs may induce cellular migration in this tumor cell line. Similarly, when MCF-7 and AGS cells were treated with Magnetic Core Mesoporous Silica nanoparticles, it increased the cell migration (58% and 53%, respectively) when compared to control group (Fig. 11 C and F)”…
Materials and Methods: Please add the source of the used antibodies.
Answer. We thank the comment made by the Reviewer. Modifications were made in the revised version.
Minor points:
The English needs to be corrected by a native speaker. There are several spelling errors (citotoxicity instead of cytotoxicity etc.), grammar mistakes (“we did not observed” instead of “we did not observe”, the word “it” as in “it is” is missing several times etc.) and general wording (“To have a better understand..” instead of “To become a better understanding…” etc.).
In the section 2.4 Intracellular signalling the authors write on several occasions: “the expression of Akt and ERK1/2 phosphorylated”. The phosphorylation of proteins is not expressed, phosphorylation is a posttranslational modification. Please rephrase.
Figure legend to figure 4: It states Tumor cells were incubated…. This is not correct, as FGH, HUVEC and NGM cells were used. It is probably a copy-paste mistake from Figure 3. Please correct.
Answer: The full text has been corrected by a native speaker (Dr. Frank Alexis). We are sorry for the previous mistakes.

Round 2
Reviewer 2 Report
In general the manuscript was revised well, the modifications in the results were well added.
Please, revise the list of references especially in the new references added.
Author Response
Reviewer #2 In general the manuscript was revised well, the modifications in the results were well added.
Please, revise the list of references especially in the new references added.
A: We would like to thank the reviewer. And also would like to inform that all the references has been revised and corrected properly